# Quantitative Analysis of Supraspinatus Tendon Pathologies via T2/T2* Mapping Techniques with 1.5 T MRI

**DOI:** 10.3390/diagnostics13152534

**Published:** 2023-07-30

**Authors:** Bunyamin Ece, Hasan Yigit, Elif Ergun, Enver Necip Koseoglu, Erdal Karavas, Sonay Aydin, Pinar Nercis Kosar

**Affiliations:** 1Department of Radiology, Kastamonu University, 37150 Kastamonu, Turkey; 2Department of Radiology, Health Sciences University, Ankara Education and Research Hospital, 06100 Ankara, Turkey; hasan.yigit@sbu.edu.tr (H.Y.); elifergun72@gmail.com (E.E.); envernecip@yahoo.com (E.N.K.); pkosar@hotmail.com (P.N.K.); 3Department of Radiology, Bandırma Onyedi Eylül University, 10200 Bandırma, Turkey; erdalkaravas@hotmail.com; 4Department of Radiology, Erzincan University, 24100 Erzincan, Turkey; sonay.aydin@erzincan.edu.tr

**Keywords:** magnetic resonance imaging, supraspinatus tendon, tendinosis, tendon rupture, T2 and T2* mapping, quantitative MRI

## Abstract

The aim of this study was to quantitatively assess supraspinatus tendon pathologies with T2/T2* mapping techniques, which are sensitive to biochemical changes. Conventional magnetic resonance imaging (MRI) and T2/T2* mapping techniques were applied to 41 patients with shoulder pathology, and there were also 20 asymptomatic cases included. The patients were divided into two groups: tendinosis and rupture. The supraspinatus tendon was divided into medial, middle, and lateral sub-regions, and the T2/T2* values were measured in both the coronal and sagittal planes for intergroup comparison. Intra-class and inter-class correlation coefficients (ICCs) were calculated to assess test reproducibility. Receiver operating characteristic (ROC) analysis was used to determine the cut-off value in each group. A total of 61 patients (27 males and 34 females)—including 20 asymptomatic individuals, 20 with tendinosis, and 21 with rupture—were evaluated using T2/T2* mapping techniques. In the rupture group, there were significant differences in the values of the lateral region (*p* < 0.001), as well as in the middle and medial regions (*p* < 0.05) of the supraspinatus tendon compared to the tendinosis and asymptomatic groups. These were determined using both T2* and T2 mapping in both the coronal and sagittal plane measurements. In the tendinosis group, there were significant differences in the values of the lateral region with T2* mapping (*p* < 0.001) in both the coronal and sagittal planes, and also with the T2 mapping in the coronal plane (*p* < 0.05) compared to the asymptomatic groups. The cut-off values for identifying supraspinatus pathology ranged from 85% to 90% for T2 measurements and above 90% for T2* measurements in both planes of the lateral section. The ICC values showed excellent reliability (ICC > 0.75) for all groups. In conclusion, T2 and T2* mapping techniques with 1.5 T MRI can be used to assess tendon rupture and tendinosis pathologies in the supraspinatus tendon. For an accurate evaluation, measurements from the lateral region in both the coronal and sagittal planes are more decisive.

## 1. Introduction

In diagnosing rotator cuff diseases, a detailed history, physical examination, and radiological imaging methods play essential roles [1]. Imaging methods are required to identify rotator cuff pathologies in most patients with shoulder pain and to determine other pathologies that may cause pain by affecting other shoulder structures. Imaging techniques such as shoulder X-ray, ultrasound, and magnetic resonance imaging (MRI) are increasingly being used for diagnostic purposes [2,3,4].

The most common cause of rotator cuff disease is the primary degeneration of the rotator cuff tendons due to aging, as well as progressive wear and tear. MRI is frequently used to diagnose and demonstrate the disease’s origin [5]. Conventional MRI plays an important role in distinguishing tendinosis from partial and full-thickness tears, determining tendon quality, detecting fatty degeneration and atrophy in muscle tissue, and recognizing other associated shoulder pathologies [6].

The degenerative process in rotator cuff tendons is characterized by alterations in the collagen structure and the biochemical composition of the tendons [5]. While advances in conventional magnetic resonance imaging (MRI) technology have allowed for the qualitative interpretation of tendon morphology, clinicians can only obtain information about the morphological condition of the tendon through a qualitative assessment of the images. However, the sensitivity and specificity of MRI may vary, particularly in cases of tendinosis and partial rotator cuff tears, and relying solely on a visual evaluation of signal intensity can lead to uncertainty in some patients [7,8,9]. Quantitative information can be clinically valuable for improving the objective interpretation of rotator cuff pathology; predicting the progression of rotator cuff disease; comparing pre- and post-treatment quantitatively; formulating treatment plans for partial rotator cuff tears and minimally retracted full-thickness tears; and for evaluating treatment outcomes. Therefore, the quantitative evaluation of tendon signal intensity is believed to increase the reliability of MRI [5,10,11,12].

Quantitative MRI techniques have recently been developed for detecting biochemical changes that occur due to degeneration in cartilage tissue and tendons [6,13,14,15,16]. Among these techniques, T2 and T2* mapping are commonly used to assess tendon pathologies. In the literature, various studies have been conducted using T2 or T2* mapping techniques in the knee [17,18], shoulder [19], hip joint cartilage [20], myocardial tissue [21], supraspinatus tendon [5,6,22], peroneal tendon [23], and Achilles tendon [16]. The results of these studies show a high potential for detecting tendinopathy with quantitative MRI.

To our knowledge, there are limited studies in the literature evaluating mapping techniques for supraspinatus tendon pathologies in shoulder joints, and all of these studies have used 3 T MRI systems [5,6,22]. This study aims to evaluate the efficacy of mapping techniques in the supraspinatus tendon when using 1.5 T MRI devices. Our objectives include utilizing T2 and T2* mapping techniques in addition to conventional MRI to quantitatively evaluate supraspinatus tendon pathologies with 1.5 T MRI, as well as to investigate the differences, if there are any, in comparison to asymptomatic individuals. We also aim to explore the possibility of quantitative and objective evaluation for tendinosis and tendon rupture pathologies by obtaining numerical values from the supraspinatus tendon.

## 2. Materials and Methods

Written informed consent was obtained from each participant for this prospective study, and the study protocol was approved by the Ethics Committee. This study was conducted following the principles of the Declaration of Helsinki.

### 2.1. Participants

The study population was planned to be divided into three groups based on supraspinatus tendon pathologies: asymptomatic, tendinosis, and tendon rupture.

To select patients for the tendinosis and rupture groups, symptomatic patients who visited our clinic’s MRI unit for conventional MRI examination with a preliminary diagnoses of rotator cuff diseases and supraspinatus tendinopathy were evaluated by clinicians (orthopedic or physical medicine and rehabilitation specialist doctors). Those who volunteered were included in the study. In the selection of tendinosis and rupture cases, those symptomatic cases that did not have previous shoulder surgery or shoulder trauma (fracture, shoulder dislocation, etc.) were included in the study. Tendinosis and tendon rupture cases were finally diagnosed via conventional MRI and grouped according to conventional MRI findings.

For the inclusion of asymptomatic cases before MRI, a physical examination was performed, and the patients were questioned as to whether they had shoulder trauma or shoulder surgery, any shoulder pain experienced during shoulder joint movements or rest, the presence of other symptoms in rotator cuff diseases, and whether there were any complaints about the shoulder. Cases without any positive findings were included in the study. Exclusion criteria included being in the pediatric age group (younger than 18 years) or the geriatric age group (older than 65 years), shoulder surgery or trauma, the presence of foreign bodies in the shoulder area, or other contraindications for MRI.

In the conventional MRI evaluation, signal changes consistent with tendinosis were detected in the images of two cases in the asymptomatic group. Thus, these patients were excluded from the study. In the pathology groups, three patients had motion artifacts, two had calcific tendinitis, and one had a full-thickness rupture with advanced retraction and superior humeral head displacement. Signal measurements in these patients were predicted to be unreliable; hence, they were excluded from the study. Therefore, finally, 20 asymptomatic cases, 20 patients in the tendinosis group, and 21 patients in the rupture group were included in the study (Figure 1).

The patients’ demographic data, such as age and gender, were recorded. Mapping measurements were performed and recorded for each patient from the medial, middle, and lateral regions of the supraspinatus tendon, including T2 coronal/sagittal and T2* coronal/sagittal mapping, resulting in a total of 12 mapping measurements. No distinction was made between the dominant and non-dominant extremities that were included in the cases.

### 2.2. Imaging Protocol

All study groups (*n* = 61) underwent conventional MRI and T2 and T2* mapping techniques, which were conducted with a 1.5 T MRI device (Magnetom Aera, Siemens Healthcare GmbH, Erlangen, Germany) using a 16-channel superficial small shoulder coil. The participants were positioned in a supine position with their palms facing upward during the examination. The imaging protocol consisted of both conventional MRI sequences and T2 and T2* mapping sequences. Details of the imaging parameters are provided in Table 1.

### 2.3. Image Analysis

The T2 and T2* mapping images were reviewed on the workstation console (Advantage Windows, GE) with rainbow color presetting, and the best view of the supraspinatus tendon was selected for the coronal sections. To assess the variability in the mapping values, as based on the anatomical location, the supraspinatus tendon was divided into three equal parts, starting from the termination level of the supraspinatus tendon in the most lateral part of the humeral head and ending at the most medial part of the humeral head. These three parts were named as the medial, middle, and lateral supraspinatus regions (Figure 2).

In the coronal sections, the round region of interest (ROI) was placed in an area of approximately 15 to 20 mm^2^ from these three sub-regions. From the localizations corresponding to the sagittal section images of the localizations measured by ROI, the measurement was conducted using a circular-shaped ROI in the sagittal sections (Figure 3). For each group, the T2 and T2* values were calculated and recorded in both the coronal and sagittal images (Figure 4 and Figure 5).

The measurements were conducted independently and at different times by a radiologist with four years of experience and a musculoskeletal radiologist with 32 years of experience. To determine the intra-rater reliability, measurements were repeated at least four to six weeks later.

### 2.4. Statistical Analysis

Statistical analysis was performed using SPSS version 15.0 software (SPSS Inc., Chicago, IL, USA). Descriptive data were expressed as the mean ± standard deviation (SD), median (min–max), or the number and frequency, as appropriate. The normal distribution of the numerical values was determined using the Kolmogorov–Smirnov and Shapiro–Wilk tests. The Student’s t-test was used for the pair group comparisons with normal distribution, while the Mann–Whitney U test was used for non-normal distribution results. The Kruskal–Wallis variance analysis was used to compare more than two groups (asymptomatic, tendinosis, and rupture groups). The Wilcoxon test was used to compare the results of two consecutive measurements in the same patient group. To investigate whether age had an effect on the MRI data, a one-way analysis of variance (ANOVA) was performed with age as a covariate. The reliability analysis test was applied to determine the intra-rater reliability between two measurements made by the same physician, and the inter-rater reliability was used for the measurements conducted between two different physicians. Intra-class and inter-class correlation coefficients (ICCs) were calculated for each region, with an ICC of 0.75 to 1.0 being considered excellent, and an ICC of 0.40 to 0.75 being considered quite good [24]. Receiver operating characteristic (ROC) analysis was used to determine the diagnostic cut-off values for tendinosis and rupture, as well as to determine the sensitivity and specificity percentages of these values for diagnosis. A *p*-value of < 0.05 was considered statistically significant.

## 3. Results

The study included a total of 61 participants, comprising 27 males and 34 females. The asymptomatic group consisted of 20 healthy individuals, the tendinosis group included 20 patients, and the tendon rupture group included 21 patients. There was no statistically significant difference in the gender distribution among the three groups (asymptomatic, tendinosis, and tendon rupture groups) (*p* = 0.262) (Table 2).

The average age of the males was 42.0 ± 11.9 years, and the average age of the females was 42.9 ± 11.9 years. There was no statistically significant difference in the mean age between the males and females in the overall study population (*p* = 0.773), nor within each of the three groups (Asymptomatic group, *p* = 0.112; Tendinosis group, *p* = 0.179; and Rupture group, *p* = 0.349) (Table 3). There was a statistically significant difference in the mean age between the three groups (*p* < 0.001). In the pairwise group comparisons, statistically significant differences in mean age were found between the asymptomatic tendinosis group (*p* = 0.023), asymptomatic tendon rupture group (*p* < 0.001), and tendinosis tendon rupture group (*p* < 0.001) (Table 3) (Figure 6). As a result of observing the statistically significant differences in age among the groups, one-way ANOVA (with age as a covariate) was performed, whereby all MR measurements from each region were used to investigate the effect of age on the group differences. The results showed that age did not have a significant effect on any of the MR measurements (*p* > 0.05) for all regions. Furthermore, when individual correlation analyses were conducted between age and each measured MR parameter, no statistically significant correlations were observed between age, nor with any of the MR parameters (*p* > 0.05).

Of the 21 patients in the rupture group, 14 had partial ruptures, and 7 had full-thickness ruptures. The rupture sizes were smaller than 1 cm in all partial ruptures and 4 cm for the full-thickness ruptures. The remaining three patients had rupture sizes between 1–2 cm. None of the patients had rupture sizes larger than 2 cm. Due to there being an insufficient number of patients for a proper grouping based on rupture size, a separate classification and comparison with statistical analysis could not be performed.

The ICC analysis revealed a good reliability for the coronal T2 measurements in the medial and middle regions, with ICC values of 0.70 and 0.71, respectively. These values are within the acceptable range for good reliability (fair to good [24]; 0.4 < ICC < 0.75). Excellent reliability was observed in all other measurements, with ICC values exceeding 0.75 (Table 4).

The MRI measurements in each region were compared among the different groups and classifications (Table 5). In the comparisons between the asymptomatic and tendinosis groups, significant differences were observed in the coronal sagittal T2* and coronal T2 measurements (*p* < 0.05) except for the sagittal T2 measurement in the lateral region (*p* > 0.05). There was no statistically significant difference in the measurements of the middle section (*p* > 0.05). However, in the medial region, significant differences were found in the coronal and sagittal T2 values (*p* < 0.05), while no significant difference was found in the coronal and sagittal T2* values (*p* > 0.05) (Table 3). In the comparisons between the asymptomatic and tendon rupture groups, the tendon rupture group was found to have statistically significant higher values in all measurements (*p* < 0.05) except for the sagittal T2* medial measurement (*p* = 0.078) (Table 5). In the comparison between the tendinosis and tendon rupture groups, statistically significant differences were observed in all measurements in the lateral region (*p* < 0.001). In the middle region, there were statistically significant differences between the two groups in all measurements (*p* < 0.05), except for the sagittal T2 measurement (*p* > 0.05). In the medial region, no statistically significant difference was found between the tendinosis and tendon rupture groups (*p* < 0.05) (Table 5).

According to the ROC analysis, there was no significant cut-off value to determine tendinosis, and the area under the ROC curve for tendinosis was not at a statistically significant level as it varied between 0.378 and 0.539. However, in the tendon rupture group, the cut-off values with the highest sensitivity and specificity, i.e., around 90% (according to the ROC analysis), were obtained from the lateral region. These findings are presented in Figure 6 and Table 6.

## 4. Discussion

Our study is a quantitative 1.5 T MRI investigation that measures, in milliseconds, the T2 and T2* times of the different anatomical subregions of the supraspinatus tendon; in addition, individuals with tendinosis or rupture to asymptomatic healthy subjects were compared, and the differences between the groups were also evaluated.

Before any measurement or assessment tool is used in research or clinical practice, its reliability needs to be investigated [25]. In our study, similar to previous studies, the intra-rater and inter-rater correlation coefficients were calculated, and excellent reliability rates were reported in most measurements, except for a few measurements that showed good reliability [5,6,22]. Neither our study nor previous studies have identified poor reliability rates [5,6,22].

There are a limited number of studies in the literature that have conducted mapping studies on the supraspinatus tendon; in these studies, Anz et al. [5] and Ganal et al. [6] divided the supraspinatus tendon into three anatomical regions and determined the boundaries of the supraspinatus muscle in each region via manual drawing techniques. They then conducted measurements, via the manual drawing technique, after creating two different groups by separating the muscles and tendons together in the first group, and only the tendon from the muscle tissue in the second group. In our study, we examined three anatomical subregions by using a circular ROI of an approximately 15–20 mm^2^ area, and focused on the central section of the tendon in both planes. This is because we used a 1.5 Tesla MRI device with a lower magnetic field strength in our study, and the resulting images may be less favorable for manual segmentation. We thought there could be more errors when using manual segmentation than in 3 T MRI devices, especially during the separation of tendinous segments from the muscle–tendon unit. In another important study that evaluated mapping in the supraspinatus tendon, Krepkin et al. [22] performed measurements using a circular ROI method like ours. Although the circular ROI method is more easily applicable, it is user-dependent and therefore increases subjectivity. In addition, the ROI method may cause variability within and between different evaluators [26]. Furthermore, repeating the ROI area of 15–20 mm^2^ in the same locations for different measurements is inevitably difficult. However, our study showed excellent intra-rater and inter-rater reliability, indicating that using a circular ROI method did not affect the results.

The most significant result of our study was the mapping data obtained from the lateral region in the tendon rupture group. According to the MR measurement values obtained in our study, it was observed that—in the evaluation of tendon ruptures compared to the other two groups (asymptomatic and tendinosis groups)—the lateral section showed the tendon rupture best in both the coronal and sagittal planes with both T2 and T2* imaging. Similar to our study, Ganal et al. [6] reported that the lateral section showed the tendon rupture best in the coronal plane with T2 mapping. However, Ganal et al. [6] also reported that the measurements obtained in the sagittal plane were not significant for showing a tendon rupture. Therefore, they concluded that the coronal plane would be more appropriate for detecting a tendon rupture. This difference in results could be attributed to the fact that the aforementioned authors used the manual drawing technique instead of the circular ROI method as a signal measurement method. With the manual drawing technique, they might have made measurements with a larger longitudinal volumetric area in the coronal plane and with a more transverse area in the sagittal section of the tendon in the sagittal plane; thus, they might not have measured the same location. In our study, the ROI area was more restricted with the round ROI placed on the tendon, and the measurements were made from the tendon areas in the sagittal plane projected with the coronal plane. In this way, the signal changes that can be caused by bursae, joint fluid, and bone structures around the joint were minimized in our study. Similarly, in the relevant studies in the literature on a similar topic, previous studies by Anz et al. [5] and Ganal et al. [6] have indicated difficulties in making measurements in the lateral section at the level of tendon termination due to partial volume effects and increased curvature of the tendon in sagittal plane images. The literature has also noted that distinguishing between small tears and tendinosis in the lateral region is particularly difficult due to an increase in partial volume effect, susceptibility to motion artifacts, the oblique–convex course, a lack of clear anatomical differentiation from the distal infraspinatus tendon, and other factors that can lead to false positive or false negative results [27,28,29,30]. Despite these difficulties, in our study, both T2 and T2* values in the sagittal plane were significantly higher in the rupture group. Thus, we observed that mapping techniques have the ability to detect different signal changes in the sagittal plane.

In our study, we found significant statistical differences in the coronal T2* and T2 measurements in both planes—not only in the lateral region, but also in the middle and medial regions—when comparing the tendon rupture group with the tendinosis group. These findings suggest that tendon rupture is not only characterized by signal changes in the lateral section, but also that it affects the data from the middle and medial sections. There could be several reasons for this, but the most likely explanation is that there are more biochemical changes in the tendon that are secondary to degeneration in patients with tendon rupture compared to those with tendinosis and who are asymptomatic. It has been reported in the literature that the rupture size detected by ultrasound and conventional MRI may be smaller than the rupture size measured during arthroscopic surgery [31]. The reason as to why the tear size detected during arthroscopy is larger than that seen in conventional MRI may be due to the inadequate visualization of the middle and medial sections, which is where the tear actually extends in conventional MRI. Our detection of significant changes in the middle and medial regions in ruptured patients using mapping methods suggests that the mapping method is successful in revealing the changes in the middle and medial regions that are inadequately shown in conventional MRI. It has been reported in the literature that the remaining fibers of the tendon in patients with tendon rupture are exposed to more biomechanical force and are more susceptible to tendinopathy. In our study, the significant signal changes detected in the middle of the tendon in the rupture group may be due to the possibility of tendon injury extending to this area and accompanying tendon degeneration and edema. In addition, fluid densities associated with the tear extend toward the middle and medial regions along the tendon sheath, and these may have affected the T2 and T2* measurements in these regions. Furthermore, studies have shown signal-enhancing fatty atrophic changes in the supraspinatus muscle in patients with chronic tendon rupture [32,33,34].

In our study, the T2* values in the coronal and sagittal planes were found to be significantly higher in the tendinosis group compared to the asymptomatic group in the lateral region. In T2 mapping, while there was no difference in the sagittal plane, the difference was significant in the coronal plane between the tendinosis and asymptomatic groups. However, the level of significance was lower than that of the T2* imaging results. Since we could not find a study in the literature that evaluates both T2* and T2 mapping together in patients with supraspinatus tendinosis, we were unable to perform a comparative analysis. According to the results obtained from our study, T2* imaging is more effective in distinguishing tendinosis from the asymptomatic group in the lateral region of the supraspinatus tendon. In our study, no significant differences were found in the T2* measurements in the medial section of the tendon in both planes when evaluating tendinosis compared to the asymptomatic group. Significantly higher results were obtained in both planes in the medial section compared to the asymptomatic group with T2 measurements. According to this result, it is understood that there is a T2 signal increase indicating biochemical changes such as edema or fatty degenerative-atrophic changes at the muscle–tendon level in both the distal and proximal sections of the tendon in tendinosis pathology. In the study by Ganal et al. [6], who evaluated T2 mapping in the supraspinatus tendon, there were, unexpectedly, statistically significant low T2 values found in the tendinosis group compared to the asymptomatic group in the medial region. In our study, the opposite result was obtained. Based on these results, it may be considered that detecting tendinosis with T2 measurements in the medial region may not be accurate.

In the literature, there are studies evaluating the fatty degeneration in the rotator cuff muscles using conventional MRI and quantitative MRI [32,33]. Goutallier et al. originally described the quality of the rotator cuff muscles on CT scans, and Fuchs et al. translated the grading to MRI [35,36]. The Goutallier semiquantitative grading system assesses the degree of fatty infiltration in the rotator cuff muscles. The grades are as follows: 0, normal muscle; I, streaks of fat; II, less fat than muscle; III, equal fat and muscle; and IV, more fat than muscle. After a rotator cuff tendon tear, the musculotendinous unit retracts permanently, undergoes atrophy and fatty infiltration, and loses elasticity [37]. In the case of a partial or full-thickness tear, changes in mapping values can occur due to fatty degeneration of the musculotendinous unit. Matsuki et al. [32], in their T2 mapping study that evaluated 101 patients with tears and 83 patients without tears for fatty degeneration, found significantly higher T2 mapping values in the supraspinatus and infraspinatus muscles in the tear group. Iijima et al. [33], in their study that evaluated fatty degeneration in rotator cuff muscles using the T2 mapping measurements from 675 patients, obtained higher T2 mapping values in patients with tears compared to those without tears, which they considered as indicative of fatty atrophic degeneration. These studies in the literature examined and included all muscle tissues and focused on the diagnosis of fatty degeneration. In our study, however, we focused more on measurements at the tendon level by using a circular ROI. Although we assume that we reduced the impact of fatty degenerative changes in the muscle, it cannot be completely ruled out that there may be some influence of fatty atrophic changes on our T2 mapping measurements in the rupture group. Prospective longitudinal studies excluding patients with fatty degenerative changes may provide additional benefits in this regard.

Tendon tissue generally has lower cell and blood vessel density, as well as metabolic activity compared to other tissues, thus making tendon healing challenging. The supraspinatus tendon is particularly vulnerable to injury and is characterized by poor blood flow, especially in the critical region located 1.5 cm proximally to the greater tubercle of the humerus. Due to these reasons, once the supraspinatus tendon is damaged, it cannot easily undergo natural repair. During tendon healing, the tissue undergoes inflammatory, repair, and remodeling phases, leading to various biochemical changes. In the inflammatory phase, a blood clot forms, attracting inflammatory cells such as pro-inflammatory cytokines, neutrophils, and macrophages to the injury site. Neovascularization occurs within the wound, supporting tissue repair. During the reparative phase, fibroblasts are the most common cell type and actively create disorganized tissue in the injury site. Type 3 collagen levels are highest at this stage and assist in collagen synthesis, gradually transforming into type 1 collagen, leading to an increased ratio of type 1 to type 3 collagen. The remodeling phase, which begins 1–2 months after the injury and lasts for over a year, is primarily characterized by type 1 collagen deposition. However, the repaired tissue cannot fully heal due to increased water content and decreased levels of collagen quantity and quality, resulting in scar tissue formation and irreversible structural changes in rotator cuff tendons. Existing data on local biochemical differences in tendinopathic tendons indicate the altered expression of collagen, proteoglycans, and matrix metalloproteinases. Additionally, cytokines such as VEGF and fibronectin have been shown to have significantly different levels in tendinopathic regions. MRI signal changes that are secondary to all these biochemical compositions and collagen structure changes in tissues provide a basis for evaluation by obtaining numerical/quantitative data with quantitative MRI mapping [38,39,40].

We conducted ROC analyses as our sample size was larger than other studies. In the ROC analysis conducted for tendinosis risk, none of the measurements provided a significant cut-off value that fell within the area under the ROC curve. The reason for this could be that the mean values and standard deviation of the tendinosis group’s measurements were quite close to those of the asymptomatic group. In the tendon rupture group, according to the ROC analysis, we obtained cut-off values with the highest sensitivity and specificity—approximately in the 90% range—from the lateral region. As ROC analysis was not conducted in previous studies, we were unable to compare our ROC analysis results with previous study results. Our study is the first in the literature to provide a cut-off value for supraspinatus tendon rupture. Further studies with larger populations are needed to determine the usability of the cut-off values in this regard.

Our study was performed with a 1.5 Tesla MR device for both T2 and T2* mapping of the supraspinatus tendon. There are studies in the literature that evaluate fatty degeneration in the supraspinatus and infraspinatus muscles with 1.5 T devices [32,33]. However, in our literature review, we could find only three studies that mapped the supraspinatus tendon [5,6,22], and 3 T MRI devices were used in these studies. To our knowledge, our study is the first to evaluate both the T2 and T2* mapping techniques in the supraspinatus tendon with 1.5 T MR devices.

In the literature, there are studies that show the differences in the measured quantitative values in mapping techniques that are based on the design of sequences and changes in the magnetic field strength. The possible reasons for these differences can be listed as follows: differences in the pulse sequence design; differences in the spatial resolution, with lower resolution being associated with more partial volume and potentially higher T2 values; and T1 relaxation effects due to higher T1 values at 3 T versus 1.5 T [41,42]. However, if we attempt to make a comparison; then, in the literature, Anz et al. [5] conducted a reference study on healthy individuals using a 3 T MRI device and shared their T2 map ms values graphically, thus making it impossible to obtain precise numerical data. In our asymptomatic group, the mean age was consistent with their younger and middle-age groups. While comparing our results with Anz et al.’s study may not be entirely accurate, it is possible to consider that our measurements were slightly lower than theirs. In another 3 T MR study that evaluated mapping in the supraspinatus tendon, Ganal et al. [6] also presented their results graphically. When compared with our study, some values were close, but it can be said that the values in our study were slightly lower. However, it should be noted that the differences in defining the region of interest in these two studies, and the inclusion of more muscle tissue in the region of interest, could have influenced the T2 mapping data, and these factors should be taken into account when making comparisons. In a study similar to ours, Krepkin et al. [22] evaluated T2, T2*, and elastography in the supraspinatus tendon by using a 3 T MRI device with a circular ROI. However, they performed anatomical sub-segmentation differently and assessed only the coronal plane. The anatomical regions they referred to as “medial” and “middle” in their study correspond to the “middle” localization in our study. The “lateral” region partially matches between the two studies. Although the number of patients in their study was relatively small, most of them were consistent with our rupture group. Therefore, when compared with our rupture group, higher values were obtained in the middle region and lower values were obtained in the lateral region in comparison to our study’s results. In this regard, there is a need for prospective studies aiming to standardize sequence parameters and measurement methods in devices with the same magnetic field strength. Although there are limitations in comparing the measurements obtained with mapping techniques across different devices, we believe that centers planning to use mapping routinely can develop and evaluate their own standardization by conducting a preliminary study on their own device.

There are some limitations to this study. A limitation of our study was the use of conventional MRI as the reference method when dividing the study population into three groups. There are many studies in the literature showing that conventional MRI has variable sensitivity and specificity in detecting rotator cuff pathologies [7,12,43]. In most studies, arthroscopy is considered the gold standard for evaluating the rotator cuff [44,45]. However, arthroscopy is an invasive method and is not routinely applicable in daily clinical practice. Moreover, it is also noted that arthroscopy is unnecessary for patients with normal tendons, tendinosis, and minimal partial tears who will not undergo surgical treatment [46,47]. Another limitation of our study was the difficulty in distinguishing between the tendinosis and asymptomatic groups when grouping the patients. There is a small possibility that there might be tendinosis patients among the cases that were considered asymptomatic. Additionally, there are various misleading factors, such as the magic angle artifact [48,49], in the recognition of tendinosis by conventional MRI. To reduce this limitation in our study, a detailed inquiry was made regarding supraspinatus tendon pathologies when determining the asymptomatic group. Furthermore, individuals with suspected disease in their history, clinical examinations, and conventional MRI were not included in the asymptomatic group. Another limitation of our study was that it was performed at a single center and using a single MRI device. In mapping studies, there may be variability in the measured quantitative values between different devices due to variations in sequence design and changes in the magnetic field strength. However, in our study, we performed imaging with the same device and sequence parameters for both pathological groups and asymptomatic cases in order to eliminate this potential error risk. In this way, we ensured optimization. However, due to the differences in sequence parameters in other studies conducted with 3 Tesla MR devices in the literature, we could not directly compare our MR measurement values. Consequently, the reproducibility of the cut-off values obtained from the tendon rupture and tendinosis groups in other devices could not be evaluated. In this regard, more studies are needed to evaluate T2/T2* mapping techniques with 1.5 Tesla MR devices and similar sequence parameters.

## 5. Conclusions

In conclusion, based on our study results, T2 and T2* mapping techniques with a 1.5 T MR device can be used for the evaluation of supraspinatus tendon pathologies. This allows for the quantitative diagnosis of tendinosis and tendon rupture cases using mapping methods, and quantitative differentiation between these two pathological groups may be possible. The T2 and T2* mapping techniques have high inter- and intra-observer reliability coefficients for evaluating the supraspinatus tendon.

## Figures and Tables

**Figure 1 diagnostics-13-02534-f001:**
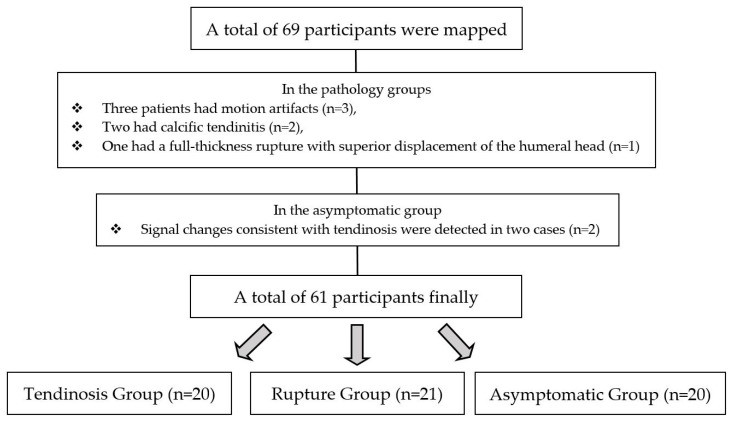
Grouping of participants and treatment methods.

**Figure 2 diagnostics-13-02534-f002:**
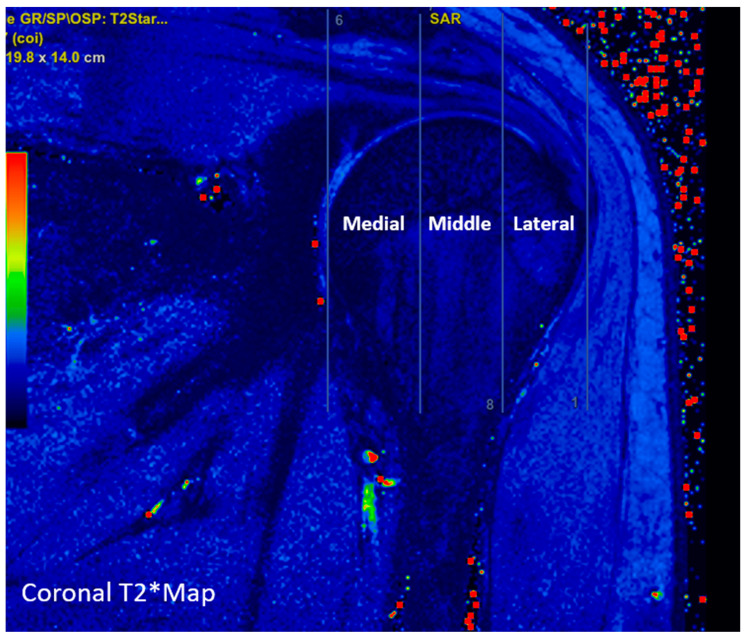
The coronal T2* image demonstrates the division of the supraspinatus tendon into three distinct anatomical subregions according to the humeral head: medial, middle, and lateral.

**Figure 3 diagnostics-13-02534-f003:**
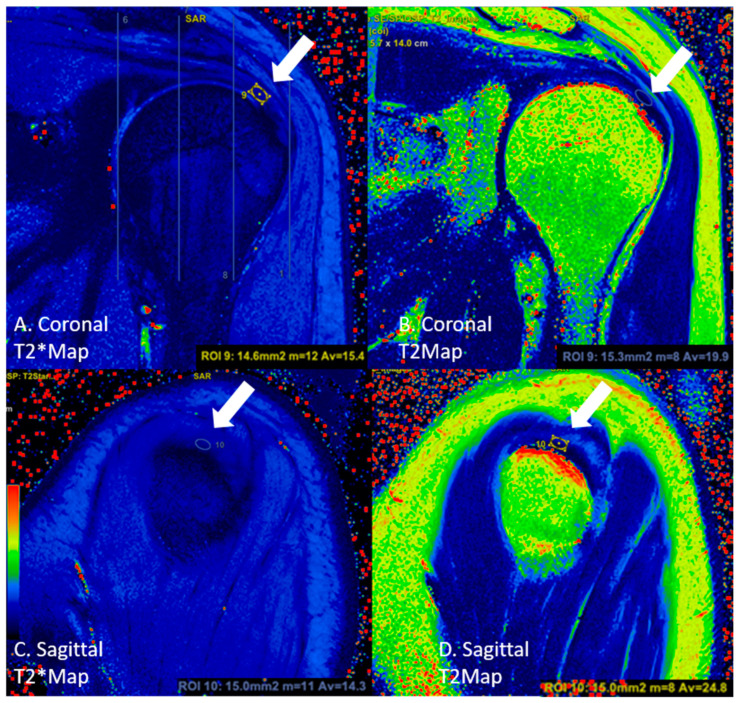
An asymptomatic case, where measurements were performed with circular ROIs (white arrows, circle) on the tendon in the lateral subregion in (**A**) coronal T2* mapping, (**B**) coronal T2 mapping, (**C**) sagittal T2* mapping, and (**D**) sagittal T2 mapping images.

**Figure 4 diagnostics-13-02534-f004:**
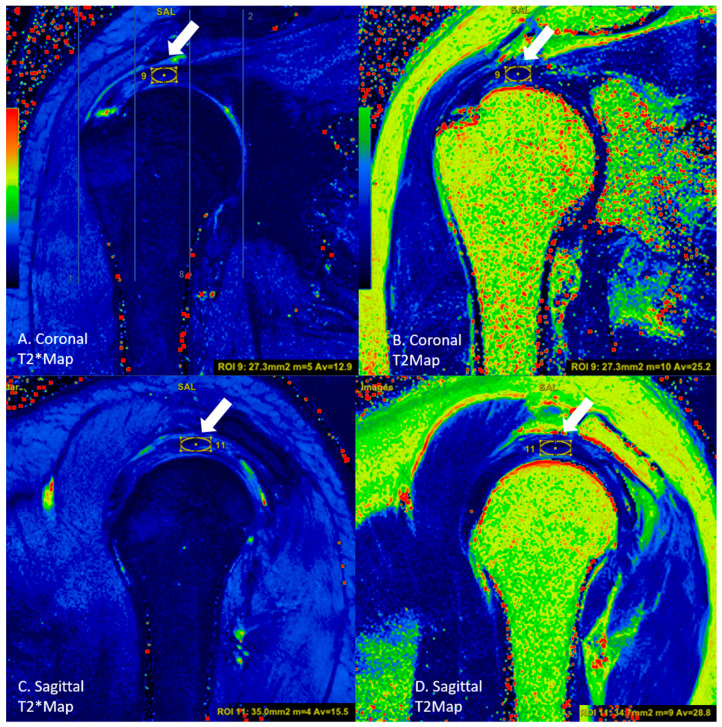
A rupture case, where measurements were performed with circular ROIs (white arrows, circle) on the tendon in the middle subregion in (**A**) coronal T2* mapping, (**B**) coronal T2 mapping, (**C**) sagittal T2* mapping, and (**D**) sagittal T2 mapping images.

**Figure 5 diagnostics-13-02534-f005:**
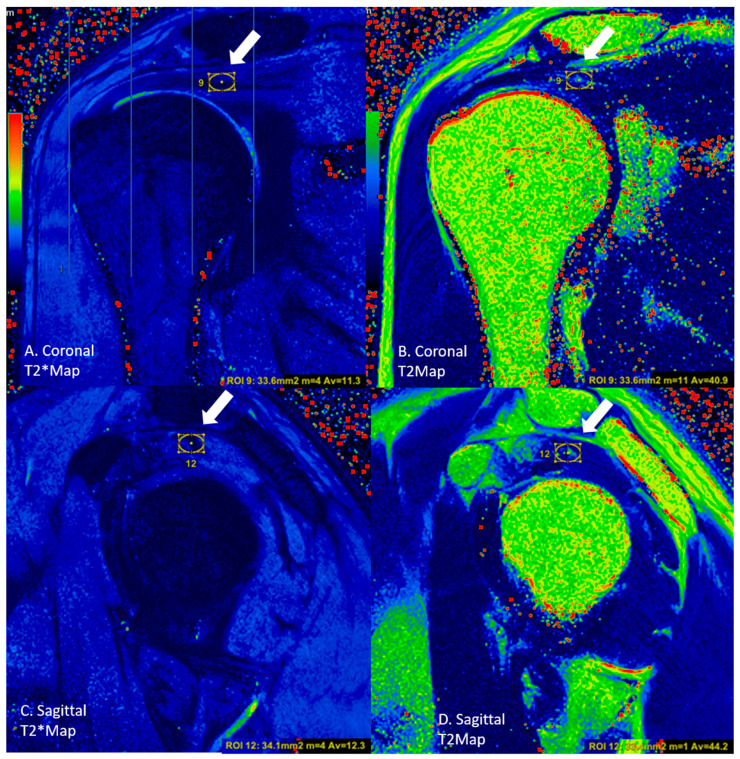
A tendinosis case, where measurements were performed with circular ROIs (white arrows, circle) on the tendon in the medial subregion in (**A**) coronal T2* mapping, (**B**) coronal T2 mapping, (**C**) sagittal T2* mapping, and (**D**) sagittal T2 mapping images.

**Figure 6 diagnostics-13-02534-f006:**
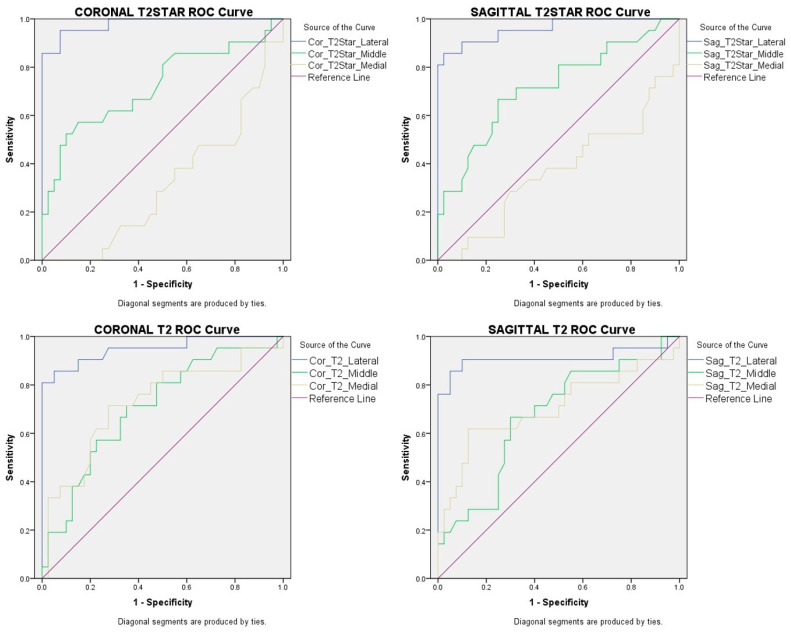
ROC analysis graph for the coronal and sagittal T2 and T2* measurements for rupture risk.

**Table 1 diagnostics-13-02534-t001:** Magnetic Resonance Imaging Parameters.

Sequence									
Parameter	Coronal T1	Sagittal T1	Coronal PD	Sagittal T2	Axial PD	Coronal T2* Map	Sagittal T2* Map	Coronal T2 Map	Sagittal T2 Map
Repetition Time (TR)	379	379	2200	3100	670	350	480	700	985
Echo Time (TE)	9.6	9.6	26	57	19	4.18 11.32 18.46 25.60 32.74	4.18 11.32 18.46 25.60 32.74	13.8 27.6 41.4 55.2 69.0	13.8 27.6 41.4 55.2 69.0
Flip Angle	150	150	180	180	28	60	60	180	180
Field of View (mm^2^)	180 mm 100%	180 mm 100%	160 mm 100%	180 mm 100%	170 mm 100%	140 mm 100%	140 mm 100%	140 mm 100%	140 mm 100%
Slice Thickness (mm)	3	4	4	4	4	3.0	3.5	3.0	3.5
Matrix	384 × 269	320 × 240	384 × 276	256 × 230	320 × 224	256 × 256	256 × 256	256 × 256	256 × 256
Acquisition Time	1:38	1:38	2:40	1:25	2:02	3:14	3:31	3:26	3:53

**Table 2 diagnostics-13-02534-t002:** Distribution of the male and female participants among the groups.

	Asymptomatic Group (*n* = 20) *n* (%)	Tendinosis Group (*n* = 20) *n* (%)	Tendon Rupture Group (*n* = 21) *n* (%)	* *p*
**Male**	11 (55.0)	6 (30.0)	10 (47.6)	0.262
**Female**	9 (45.0)	14 (70.0)	11 (52.4)	

* Chi-Square Test.

**Table 3 diagnostics-13-02534-t003:** Comparison of the mean age between male and female participants in all three groups.

	Asymptomatic Group (*n* = 20)		Tendinosis Group (*n* = 20)		Tendon Rupture Group (*n* = 21)		*p* Value
Age (Years) (Mean ± SD)	33.2 ± 8.1		40.8 ± 10.3		52.9 ± 7.4		**<0.001**
	**Male**	**Female**	**Male**	**Female**	**Male**	**Female**	
Age (Years) (Mean ± SD)	31.1 ± 7.4	35.8 ± 8.6	45.7 ±8.4	38.7 ±10.6	51.7 ±7.2	53.9 ±7.8	
*p* Value	0.112		0.179		0.349		

**Table 4 diagnostics-13-02534-t004:** Inter-rater and intra-rater reliability (95% confidence interval) for the T2* and T2 mapping differences of anatomical regions according to imaging planes.

Section	Region	Inter-Rater Reliability	Intra-Rater Reliability
Coronal	Lateral		
	T2*	0.968 (0.947–0.981)	0.984 (0.973–0.990)
	T2	0.965 (0.942–0.979)	0.986 (0.976–0.991)
	Middle		
	T2*	0.906 (0.843–0.944)	0.979 (0.965–0.987)
	T2	0.702 (0.503–0.821)	0.934 (0.891–0.961)
	Medial		
	T2*	0.781 (0.635–0.868)	0.832 (0.720–0.899)
	T2	0.838 (0.730–0.903)	0.717 (0.528–0.830)
Sagittal	Lateral		
	T2*	0.962 (0.937–0.977)	0.981 (0.969–0.989)
	T2	0.954 (0.924–0.973)	0.993 (0.988–0.996)
	Middle		
	T2*	0.909 (0.849–0.945)	0.969 (0.948–0.981)
	T2	0.915 (0.858–0.949)	0.918 (0.863–0.951)
	Medial		
	T2*	0.788 (0.647–0.873)	0.843 (0.738–0.906)
	T2	0.922 (0.871–0.954)	0.888 (0.814–0.933)

T2*: T2Star

**Table 5 diagnostics-13-02534-t005:** Comparisons of the T2 and T2* MRI signal values between the asymptomatic, tendinosis, and tendon rupture groups.

		Asymptomatic Group ^1^ (*n* = 20)	Tendinosis Group ^2^ (*n* = 20)	Tendon Rupture Group ^3^ (*n* = 21)	^1,2^ *p* *	^1,3^ *p* *	^2,3^ *p* *
		Mean ± SD	Mean ± SD	Mean ± SD			
Coronal T2*	Lateral	14.1 ± 3.3	18.1 ± 3.6	33.4 ± 8.3	**<0.001**	**<0.001**	**<0.001**
	Middle	12.2 ± 2.1	13.0 ± 3.0	16.4 ± 5.2	0.957	**0.007**	**0.028**
	Medial	13.1 ± 1.9	12.3 ± 2.4	11.2 ± 1.6	0.256	**0.006**	0.188
Sagittal T2*	Lateral	15.3 ± 2.6	19.1 ± 2.8	31.4 ± 7.3	**<0.001**	**<0.001**	**0.001**
	Middle	12.6 ± 2.3	13.2 ± 3.2	16.2 ± 4.6	0.850	**0.011**	**0.027**
	Medial	13.2 ± 1.4	13.2 ± 2.3	12.2 ± 1.9	0.534	0.078	0.268
Coronal T2	Lateral	21.7 ± 3.1	24.6 ± 3.8	45.5 ± 12.3	**0.021**	**<0.001**	**<0.001**
	Middle	23.1 ± 3.4	23.9 ± 4.5	26.5 ± 5.0	0.957	**0.014**	**0.038**
	Medial	33.5 ± 7.3	39.9 ± 7.6	43.4 ± 9.5	**0.008**	**0.001**	0.085
Sagittal T2	Lateral	25.2 ± 3.3	28.2 ± 5.0	46.9 ± 12.8	0.064	**<0.001**	**<0.001**
	Middle	24.0 ± 4.1	25.5 ± 4.5	27.9 ± 5.4	0.137	**0.012**	0.215
	Medial	33.8 ± 5.9	38.9 ± 6.7	42.6 ± 9.8	**0.023**	**0.002**	0.155

T2*:T2Star, SD: Standard deviation, * Mann–Whitney U test.

**Table 6 diagnostics-13-02534-t006:** ROC analysis values for the coronal and sagittal T2 and T2* measurements in the tendon rupture risk analysis.

	Coronal	Sagittal	Coronal	Sagittal
	T2*Lateral	T2*Lateral	T2Lateral	T2Lateral
Area Under the ROC Curve (95% Confidence Interval)	0.980 (0–1)	0.960 (0–1)	0.949 (0–1)	0.911 (0–1)
Standard error	0.015	0.026	0.026	0.054
*p* Value	<0.001	<0.001	<0.001	<0.001
Cut-off Value (ms)	22.17	21.07	26.97	31.8
Sensitivity	%95.2	%90.5	%90.5	%90.5
Specificity	%92.5	%90	%85	%90

T2*: T2Star.

## Data Availability

The data that support the findings of this study are available from the corresponding authors upon reasonable request.

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
