# Peer review of "Quantitative Analysis of Supraspinatus Tendon Pathologies via T2/T2* Mapping Techniques with 1.5 T MRI"

_diagnostics, 2023, doi:10.3390/diagnostics13152534_

Round 1

Reviewer 1 Report

In tendon rupture, supraspinatus shows high intensity, which is usually diagnostic on conventional MRI. Please emphasize the significance of “mapping” in this study. In particular, please discuss the quality of the rotator cuff, since there is no discussion of rotator cuff quality.

How do the results of this study compare to previous studies that have used 3T MRI devices for T2 and T2* mapping of the supraspinatus tendon? Are there any significant differences in the results obtained using 1.5T and 3T MRI devices?

The rotator cuff degenerates with age and often shows high intensity on T2 image. Please mention the age distribution of the three groups in this study.

Similarly, could the authors provide more information on how they controlled for potential confounding factors such as age, gender, and medical history?

Please provide ‘tear size’ in the tendon rupture group.

Please describe how you diagnosed tendinitis and tendon rupture. Physical examination reveals similar symptoms; if you have differentiated them by conventional MRI findings, please describe the differentiation method.

In fig6, it is hard to understand which figure represents which MRI view. Caption is needed.

L283 Please discuss what kind of ‘biochemical changes’ occurs after tendon rapture. 

Author Response

Manuscript ID: diagnostics-2458779

Quantitative Analysis of Supraspinatus Tendon Pathologies via T2/T2* Mapping Techniques with 1.5 T MRI

Dear Reviewer,

We appreciate the time and effort that you dedicated to providing feedback on our manuscript and are grateful for the insightful comments and valuable improvements to our paper. We have made the relevant corrections according to your suggestions. Please see below for a point-by-point response to the comments and suggestions. And please see the attachment for the last edited version of the article with track changes enabled format-Word file.

1) 1a) In tendon rupture, supraspinatus shows high intensity, which is usually diagnostic on conventional MRI. Please emphasize the significance of “mapping” in this study.

1b) In particular, please discuss the quality of the rotator cuff, since there is no discussion of rotator cuff quality.

1a. Response:

In line with your suggestions, we have added some sentences to the relevant section (introduction) to further emphasize the significance of “mapping” in the study. The new version of the relevant paragraph is as follows.

“The degenerative process in rotator cuff tendons is characterized by alterations in the collagen structure and biochemical composition of the tendons [5]. While advances in conventional magnetic resonance imaging (MRI) technology have allowed for qualitative interpretation of tendon morphology, clinicians can only obtain information about the morphological condition of the tendon through qualitative assessment of the images. However, the sensitivity and specificity of MRI may vary, particularly in cases of tendinosis and partial rotator cuff tears, and relying solely on visual evaluation of signal intensity can lead to uncertainty in some patients [7-9]. Quantitative information can be clinically valuable for improving the objective interpretation of rotator cuff pathology, predicting the progression of rotator cuff disease, comparing pre- and post-treatment quantitatively, formulating treatment plans for partial rotator cuff tears and minimally retracted full-thickness tears, and evaluating treatment outcomes. Therefore, the quantitative evaluation of tendon signal intensity is believed to increase the reliability of MRI [5,10-12].”

1b. Response:

We have added a paragraph discussing the quality of the rotator cuff to the discussion section based on your suggestions.

“In the literature, there are studies evaluating fatty degeneration in the rotator cuff muscles using conventional MRI and quantitative MRI (32,33). Goutallier et al. originally described the quality of the rotator cuff muscles on CT scan, and Fuchs et al. translated the grading to MRI (35,36). The Goutallier’s semiquantitative grading system assesses the degree of fatty infiltration in the rotator cuff muscles. The grades are as follows: 0, normal muscle; I, streaks of fat; II, less fat than muscle; III, equal fat and muscle; and IV, more fat than muscle. After a tear of the rotator cuff tendon, the musculotendinous unit retracts permanently, undergoes atrophy and fatty infiltration, and loses elasticity (37). In the case of a partial or full-thickness tear, changes in mapping values can occur due to fatty degeneration of the musculotendinous unit. Matsuki et al. (32), in their T2 mapping study evaluating 101 patients with tears and 83 patients without tears for fatty degeneration, found significantly higher T2 mapping values in the supraspinatus and infraspinatus muscles in the tear group. Iijima et al. (33), in their study evaluating fatty degeneration in rotator cuff muscles using T2 mapping measurements from 675 patients, obtained higher T2 mapping values in patients with tears compared to those without tears, which they considered as indicative of fatty atrophic degeneration. These studies in the literature examined and included all muscle tissues and focused on the diagnosis of fatty degeneration. In our study, however, we focused more on measurements at the tendon level using a circular ROI. Although we assume that we reduced the impact of fatty degenerative changes in the muscle, it cannot be completely ruled out that there may be some influence of fatty atrophic changes on our T2 mapping measurements in the rupture group. Prospective longitudinal studies excluding patients with fatty degenerative changes may provide additional benefit in this regard.”

2) How do the results of this study compare to previous studies that have used 3T MRI devices for T2 and T2* mapping of the supraspinatus tendon? Are there any significant differences in the results obtained using 1.5T and 3T MRI devices?

2.Response:

In line with your suggestions, we have added the following paragraph to the discussion section.

“In the literature, there are studies showing differences in the measured quantitative values in mapping techniques based on the design of sequences and changes in the magnetic field strength. The possible reasons for these differences can be listed as follows: differences in the pulse sequence design, differences in the spatial resolution, with lower resolution being associated with more partial volume and potentially higher T2 values, and T1 relaxation effects due to higher T1 values at 3 T versus 1.5 T [42,43]. However, if we attempt to make a comparison; in the literature, ANZ et al. conducted a reference study on healthy individuals using a 3T MRI device and shared their T2 map ms values graphically, making it impossible to obtain precise numerical data. In our asymptomatic group, the mean age, which was consistent with ANZ et al.'s younger and middle-age groups. While comparing our results with ANZ et al.'s study may not be entirely accurate, it is possible to consider that our measurements were slightly lower than theirs. In another 3T MR study evaluating mapping in the supraspinatus tendon, Ganal et al. also presented their results graphically. When compared with our study, some values were close, but it can be said that the values in our study were slightly lower. However, it should be noted that the differences in defining the region of interest in these two studies, and the inclusion of more muscle tissue in the region of interest, could have influenced the T2 mapping data, and these factors should be taken into account when making comparisons. In a study similar to ours, Krepkin et al. evaluated T2, T2*, and elastography in the supraspinatus tendon using a 3T MRI device with a circular ROI. However, they performed anatomical sub-segmentation differently and assessed only the coronal plane. The anatomical regions they referred to as "medial" and "middle" in their study correspond to the "middle" localization in our study. The "lateral" region partially matches between the two studies. Although the number of patients in their study was relatively small, most of them were consistent with our rupture group. Therefore, when comparing with our rupture group, higher values were obtained in the middle region, and lower values were obtained in the lateral region compared to our study's results. In this regard, there is a need for prospective studies aiming to standardize sequence parameters and measurement methods in devices with the same magnetic field strength. Although there are limitations in comparing measurements obtained with mapping techniques across different devices, we believe that centers planning to use mapping routinely can develop and evaluate their own standardization by conducting a preliminary study on their own device.”

3) The rotator cuff degenerates with age and often shows high intensity on T2 image. Please mention the age distribution of the three groups in this study.

3.Response:

In line with your suggestions, we made some additions. The parts we added are as follows.

“The average age of males was 42.0 ± 11.9 years, and the average age of females was 42.9 ± 11.9 years. There was no statistically significant difference in the mean age between males and females in the overall study population (p=0.773) and within each of the three groups (Asymptomatic group, p=0.112; Tendinosis group, p=0.179; Rupture group, p=0.349) (Table 3).

There was a statistically significant difference in mean age among the three groups (p<0.001). In pairwise group comparisons, statistically significant differences in mean age were found between the asymptomatic-tendinosis group (p=0.023), asymptomatic-tendon rupture group (p<0.001), and tendinosis-tendon rupture group (p<0.001) (Table 3) (Figure 6). As a result of observing statistically significant differences in age among the groups, one-way ANOVA with age as a covariate was performed, using all MR measurements from each region to investigate the effect of age on group differences. The results showed that age did not have a significant effect on any of the MR measurements (p>0.05) for all regions. Furthermore, when individual correlation analyses were conducted between age and each measured MR parameter, no statistically significant correlations were observed between age and any of the MR parameters (p>0.05).”

Table 3. Comparison of mean age (mean ± standard deviation) between male and female participants in all three groups

Asymptomatic Group1

(n=20)

Tendinosis Group2

(n=20)

Tendon Rupture Group3

(n=21)

P Value

Age (Years)

33.2 ± 8.1

40.8 ± 10.3

52.9 ± 7.4

<0.001

Male

Female

Male

Female

Male

Female

Age (Years)

31.1 ± 7.4

35.8 ± 8.6

45.7 ±8.4

38.7 ±10.6

51.7 ±7.2

53.9 ±7.8

P Value

0.112

0.179

0.349

4) Similarly, could the authors provide more information on how they controlled for potential confounding factors such as age, gender, and medical history

4.Response:

We have made some additions in line with your suggestions.

“A total of 61 individuals, 27 males and 34 females, were included in the study. The asymptomatic group consisted of 20 healthy individuals, the tendinosis group included 20 patients, and the tendon rupture group included 21 patients. There was no statistically significant difference in the gender distribution among the three groups (asymptomatic, tendinosis, and tendon rupture groups) (p=0.262) (Table 2).”

Table 2. Distribution of male and female participants among the groups.

Asymptomatic Group1

(n=20)

n (%)

Tendinosis Group2

(n=20)

n (%)

Tendon Rupture Group3

(n=21)

n (%)

*P

Male

11 (55.0)

6 (30.0)

10 (47.6)

0.262

Female

9 (45.0)

14 (70.0)

11 (52.4)

*Chi-Square Test

5) Please provide ‘tear size’ in the tendon rupture group.

5.Response:

In line with your suggestions, the following sentences have been added to the results section.

“Out of the 21 patients in the rupture group, 14 had partial ruptures, and 7 had full-thickness ruptures. The rupture sizes were smaller than 1 cm in all partial ruptures and 4 of the full-thickness ruptures. The remaining 3 patients had rupture sizes between 1-2 cm. None of the patients had rupture sizes larger than 2 cm. Due to the insufficient number of patients for proper grouping based on rupture size, a separate classification and comparison with statistical analysis could not be performed.”

6) Please describe how you diagnosed tendinitis and tendon rupture. Physical examination reveals similar symptoms; if you have differentiated them by conventional MRI findings, please describe the differentiation method.

6.Response:

Yes, we categorized the patients based not only on clinical findings but also on MRI findings. However, we now realize that we did not express this clearly. Thank you for pointing that out. In line with your suggestions, we have added the following sentence to the material method section:

"Tendinosis and tendon rupture cases were finally diagnosed using conventional MRI and grouped according to conventional MRI findings."

7) In fig6, it is hard to understand which figure represents which MRI view. Caption is needed.

7.Response:

In line with your suggestions, Figure 6 was reviewed again, and captions were added, and enlarging the font size. The previous and updated versions of Figure 6 are shared below, respectively.

Old version of Fig 6 is below:

New version of Fig 6 is below:

8) L283 Please discuss what kind of ‘biochemical changes’ occurs after tendon rapture.

8.Response:

In line with your suggestions, we have added a paragraph regarding the biochemical changes that occur during tendon injury and healing.

“Tendon tissue generally has lower cell and blood vessel density and metabolic activity compared to other tissues, making tendon healing challenging. The supraspinatus tendon is particularly vulnerable to injury and is characterized by poor blood flow, especially in the critical region located 1.5 cm proximal to the greater tubercle of the humerus. Due to these reasons, once the supraspinatus tendon is damaged, it cannot easily undergo natural repair. During tendon healing, the tissue undergoes inflammatory, repair, and remodeling phases, leading to various biochemical changes. In the inflammatory phase, a blood clot forms, attracting inflammatory cells such as pro-inflammatory cytokines, neutrophils, and macrophages to the injury site. Neovascularization occurs within the wound, supporting tissue repair. During the reparative phase, fibroblasts are the most common cell type and actively create disorganized tissue in the injury site. Type 3 collagen levels are highest at this stage and assist in collagen synthesis, gradually transforming into type 1 collagen, leading to an increased ratio of type 1 to type 3 collagen. The remodeling phase, which begins 1-2 months after the injury and lasts for over a year, is primarily characterized by type 1 collagen deposition. However, the repaired tissue cannot fully heal due to increased water content and decreased levels of collagen quantity and quality, resulting in scar tissue formation and irreversible structural changes in rotator cuff tendons. Existing data on local biochemical differences in tendinopathic tendons indicate altered expression of collagen, proteoglycans, and matrix metalloproteinases. Additionally, cytokines such as VEGF and fibronectin have been shown to have significantly different levels in tendinopathic regions. MRI signal changes secondary to all these biochemical composition and collagen structure changes in tissues provide evaluation by obtaining numerical/quantitative data with quantitative MRI mapping[38–40].”

Thank you very much for all your valuable comments, suggestions, contributions and efforts.

Reviewer 2 Report

In this study, the effectiveness of T2/T2* mapping techniques with 1.5 T MRI was evaluated for the Quantitative Analysis of Supraspinatus Tendon pathologies.

What software did the authors use for mapping?

How did the authors encounter the magic angle effect in the supraspinatus tendon?

Provide some clinical applications for this mapping. Can radiologists’ diagnoses be replaced by this method?

It needs some revisions by a native English language expert.

Author Response

Manuscript ID: diagnostics-2458779

Quantitative Analysis of Supraspinatus Tendon Pathologies via T2/T2* Mapping Techniques with 1.5 T MRI

Dear Reviewer,

We appreciate the time and effort that you dedicated to providing feedback on our manuscript and are grateful for the insightful comments and valuable improvements to our paper. We have made the relevant corrections according to your suggestions. Please see below for a point-by-point response to the comments and suggestions. And please see the attachment for the last edited version of the article with track changes enabled format-Word file.

1) In this study, the effectiveness of T2/T2* mapping techniques with 1.5 T MRI was evaluated for the Quantitative Analysis of Supraspinatus Tendon pathologies.

What software did the authors use for mapping?

1.Response:

"The T2 and T2* mapping images were reviewed on the workstation console (Advantage Windows, GE) with rainbow color presetting, and the best view of the supraspinatus tendon was selected in coronal sections.”

Additionally, we did not use any external software.

2) How did the authors encounter the magic angle effect in the supraspinatus tendon?

2.Response:

In fact, due to our imaging quality and protocol, the magic angle artifact is not a very common occurrence for us. However, theoretically, there is a slight risk of the magic angle artifact, considering that we have identified asymptomatic cases and tendinosis cases using conventional MRI. Therefore, we have mentioned this limitation in the limitations section.

3) Provide some clinical applications for this mapping. Can radiologists’ diagnoses be replaced by this method?

3.Response:

In line with your suggestions, we have added the following sentences to the introduction section regarding the clinical usability of mapping.

“Quantitative information can be clinically valuable for improving the objective interpretation of rotator cuff pathology, predicting the progression of rotator cuff disease, comparing pre- and post-treatment quantitatively, formulating treatment plans for partial rotator cuff tears and minimally retracted full-thickness tears, and evaluating treatment outcomes. Therefore, the quantitative evaluation of tendon signal intensity is believed to increase the reliability of MRI [5,10-12].”

4) It needs some revisions by a native English language expert.

4.Response:

We have revised the manuscript following your suggestions, used an artificial intelligence-based application, and had a native speaker review it again. We made some revisions to improve the English language quality.

Thank you very much for all your valuable comments, suggestions, contributions, and efforts.
